# Population genetic structure and historical demography of the population of forest elephants in Côte d'Ivoire

Jean-Louis Kouakou[1], Sery Gonedelé-Bi[2,3]*

1 Laboratoire de Génomique Fonctionnelle et Amélioration Génétique, Université Nangui Abrogoua, Abidjan, Côte d'Ivoire, 2 Laboratoire de Biotechnologie, Agriculture et Valorisation des Ressources Biologiques, Université Félix Houphouët Boigny, Abidjan-Cocody, Côte d'Ivoire, 3 Centre Suisse de Recherches Scientifiques en Côte d'Ivoire, Adiopodoumé, Côte d'Ivoire

* sgonedele@gmail.com

**Data Availability Statement:** Nucleotide sequences data are available from GenBank (https://www.ncbi.nlm.nih.gov).

**Funding:** The author(s) received no specific funding for this work.

## Abstract

The population of forest elephant (*Loxodonta cyclotis*) has continuously declined in Côte d'Ivoire and, the remaining population largely consists of subpopulations that are fragmented and isolated. No data actually exist on the level of genetic diversity and population genetic structure of current forest elephant populations in Côte d'Ivoire. In this sense, determining genetic diversity and the underlying mechanisms of population differentiation is crucial for the initiation of effective conservation management. A total of 158 dung samples of forest elephants were collected at stage 1 of decompositions (dung pile intact, very fresh) in three Classified Forests (CF) (Bossématié, Dassioko and Port-Gauthier) in Côte d'Ivoire. A total of 101 sequences of the mitochondrial DNA control region measuring 600 base pair and 26 haplotypes were obtained. A haplotypic diversity ranging from 0.655 ± 0.050 at Bossématié and 0.859 ± 0.088 at Port Gauthier was obtained. Fifteen (15) out of 26 haplotypes observed were singletons and only the Dassioko and Port Gauthier CFs shared the same haplotypes. The strong genetic connectivity between forest elephant populations of the Dassioko and Port Gauthier CFs is supported by the grouping of these populations into a single cluster by Bayesian analysis. Although populations of *L. cyclotis* exhibit relatively high genetic diversity, habitat fragmentation could affect the genetic variability of current populations. Urgent measures including the reinforcement/establishment of genetic corridors and the strengthening of protection measures need to be undertaken to save the remaining populations of forest elephants in Côte d'Ivoire.

## Introduction

African forest elephants formerly appeared across the entire humid forest area of western and central Africa, and are currently found in 20 countries. Their range is decreasing and is highly fragmented in western Africa where seven range countries are reported to have fewer than a hundred African forest elephants each [1, 2]. Combined effect of increasing human population

**Competing interests:** The authors have declared that no competing interests exist.

growth, habitat loss and fragmentation and continuous poaching for ivory have decimated the population of forest elephants. These threats are particularly severe in West Africa where even protected areas that serve as refuges for these large mammals are encroached for farming in the absence of effective control measure [3–6]. The level of threats affecting forest elephants has held to the upgrade of the level of conservation status from Endangered to Critically Endangered by the International Union for the Conservation of Nature (IUCN) [2].

In Côte d'Ivoire, of an early population of elephant estimate of 3050 individuals throughout the country [6], the population of forest elephant has continuously declined. The remaining population is highly fragmented and largely live in nominally protected national parks and forest reserves. Recent population estimate by Kouakou et al. [5] indicate that these populations consist of four fragmented and isolated populations. In these subpopulations, genetic drift and increased inbreeding can reduce genetic diversity [7, 8]. The viability of the remaining population is then questionable. No data actually exist on the level of genetic diversity and population genetic structure of current forest elephant populations of Côte d'Ivoire and the genetic conservation status of this species. In this sense, determining genetic diversity and the underlying mechanisms of population differentiation in impacted populations is crucial for the initiation of effective conservation management [9].

Mitochondrial control region has been widely used for the study of population genetic structure of species because it is more variable than other region and it lacks recombination [10, 11]. In this study, we used the mitochondrial DNA (mtDNA) control region to evaluate genetic diversity, the population genetic structure, and population demographic history of forest elephant in Côte d'Ivoire. The control region of mitochondrial DNA (mtDNA), which is only transmitted maternally provides information on the maternal heredity of the individuals examined and may provide additional information on the origins of the populations studied [12]. mtDNA has in fact been shown to be extremely informative in phylogeographic studies, as it exhibits relatively rapid rates of evolution and does not undergo recombination between loci [13]. The viability of the remaining population of forest elephant, their connectivity, and the existence of any genetic signatures were determined for their effective conservation and management in Côte d'Ivoire. In particular, it was important to determine whether observed reduction in population size was accompanied by depletion at the levels of genetic diversity and also to detect genetic signatures of past demographic changes coupled with the effect of recent human induced disturbance.

## Materials and methods

### Samples collection

From 07 June to 30 October 2019, a total of 158 dung samples of forest elephants were collected at stage 1 of decompositions (dung pile intact, very fresh, and moist, with odour) in the Bossématié (53 dung samples), Dassioko (66 dung samples), Port-Gauthier (31 dung samples) Classified Forests (CF) and the Zoo of Abidjan (8 dung samples) respectively located in the rainforest area in Côte d'Ivoire. Dung samples were collected by gloved hand using single-use dry sticks to avoid all sources of contamination. Using the dry stick, the upper part of the droppings was scraped off and approximately 5 g of droppings were transferred into 50 mL cryotube containing 20% Dimethyl sulfoxide (20% DMSO) [14]. Elephant dung samples were collected during transect line surveys. The equidistance between the transect lines was 2 km. During the survey, all dung piles from which samples were collected were destroyed to avoid re-sampling from the same dung pile. Samples from dung of the same size, at the same location, on the same day were avoided to limit re-sampling the same individual. A total of 158 samples were collected. All samples were transported to the laboratory of Centre Suisse de

Recherches Scientifiques in Côte d'Ivoire (CSRS) within 15–20 days and stored at -80˚C till extraction.

Before conducting the study, we received research permits from SODEFOR (Society of Forest Development) and OIPR (Ivorian Office of Parks and Reserves), respectively in charge of the management of Côte d'Ivoire's forest reserves and parks. The study was approved by SODEFOR and OIPR.

## DNA extraction, PCR and sequencing

DNA was successfully extracted from all 158 fecal samples following the standard Cetyl Trimethyl Ammonium Bromide (CTAB) method [15] optimized by our in house protocol [16]. The spectrophotometric method was used for the quantification of total DNA extracted from dung samples. Each sample was tested three times. A volume of 1 μL of each total DNA extract was used to determine the quantity and quality of the total DNA using a Nanodrop spectrophotometer (Thermo Scientific NanoDrop 2000 C) which calculated the absorbance ratio A260nm / A280nm of each sample.

Primer pairs LafCR1 F: 5'-GTATAAGACATTACAATGGTC-3' (10μM) and LafCR2 R: 5'-AGATGTCTTATTTAAGAGGA-3' [17], were used to amplify approximately 600 base pairs (bp) of the mtDNA segment located in the hypervariable I region of the control region. Polymerase chain reaction (PCR) was performed in a total volume of 50 μL reactions containing 1 μl of genomic DNA (338,15 ± 528,54 ng/uL), 25 μl Quick-Load one Taq 2X Master Mix with Standard Buffer (New ENGLAND Biolabs, USA), 1 μL of Dimethyl sulfoxide (10%) (Eurolabs, France), 1 μL of magnesium chloride (MgCl$_2$) (New ENGLAND Biolabs, USA) (25 mM), 4 μl of Bovin serum albumin (BSA) (SIGMA-ALDRICH, Germany) (10%), 1 μL of each primer (10 μM) and 15 μL volume ultrapure water (H$_2$O). This mixture was processed in the thermalcycler Techne TC-512 for PCR reactions at different successive temperature cycles. A total of 40 cycles were performed. Each cycle consists respectively of an initial denaturation at 95˚C for 5 min, followed by denaturation at 95˚C for 30 seconds, a hybridization phase at 58.3˚C for 45 seconds and an elongation at 72˚C for 30 s, then a final termination cycle at 72˚C for 5 min and finally a cooling phase at 4˚C.

To avoid nuclear mitochondrial DNA amplification, we used primers pairs that are specific to the control region (CSB1) of the mitochondrial DNA. The mtDNA sequence from the Abidjan Zoo elephant was considered as the reference DNA sequence in this study. The validity of the sequences generated was tested by comparing them to the existing sequences using BLAST version 2.2.32 [18].

To monitor for contamination, DNA extractions and PCRs contained one to three negative controls.

DNA sequencing was performed on the two strands of DNA by BGI BIO-SOLUTIONS HONGKONG CO., LIMITED company following the techniques of Sanger et al. [19].

## Data analyses

Sequencing errors were corrected by comparing the sequencing product obtained from the sense strand of the primer during amplification with the sequencing product obtained from the anti-sense strand of the primer of each of the 101 amplicons. The ends of sequences corresponding to the annealing regions are often subject to errors, these regions have been removed. Both the forward and reverse sequences were assembled into one sequence using the overlap of the two sequences using the BioEdit version 7 software (https://itservices.cas.unt.edu/software/bioedit725).

The total 101 good quality sequences generated were aligned using the Basic Local Alignment Search Tool (BLAST) interface in the National Center for Biotechnology Information database at GenBank in order to determine similarities with the other sequences already identified as mitochondrial DNA sequences (mtDNA). We also looked at "CDS Features" and translations of all amino acids translations of all coding regions (the start and stop codons and the locations of all exons).

A total of 101 sequences of mitochondrial control region generated were aligned using the CLUSTAL X program [20] with the default settings and verified by eye. The programme DnaSP 6.11.01 [21] was used to calculate genetic diversity indices (number of segregating sites (S), number of haplotypes (Nh), haplotype diversity (h), and nucleotide diversity ($\pi$). Haplotype networks were constructed to identify genealogical relationships at the intraspecific level, as well as to make inferences about biogeography and population history using POPART version 1.7 software [22] following the Median Joining Network method [23]. Pairwise genetic differentiation using the index $\Phi$ST between sampling sites were estimated using the software ARLEQUIN version 3.5.2.2 [24] with pairwise difference as the distance method and 20,000 permutations. The software GenAIEx version 6.51b2 was also used to perform a Mantel test to evaluate a pattern of isolation by distance (IBD) taking into account the genetic distance and the geographical distance between samples [25] Tajima's (D) [26] and Fu's (Fs) [27] tests were conducted in DnaSP to assess neutrality and as a first estimate of potential population sizes changes in the past [21].

The phylogenetic tree was constructed using Unweighted pair group method with arithmetic mean (UPGMA) with the MEGA software [28] under Hasegawa–Kishono–Yano (HKY+G) model (which is the closest available MEGA model to GTR which was the best nucleotide substitution model determined with jModelTest 2.1.10 [29].

In addition to the 101 sequences generated by the present study, 11 mDNA control region nucleotide sequence (Table 1) were collected from GenBank database for comparative phylogenetic analyses.

The robustness of the phylogenetic inference was assessed using 1000 bootstrap replicates. Genetic differentiation between populations was assessed by comparing the average number of pairwise differences between populations, within populations (PiX and PiY) and the corrected average pairwise difference (PiXY(PiX þPiY)/2), respectively, using the program ARLEQUIN 3.5.2.2 [24]. The significance of differentiation between pairs of populations was tested by exact test [30] using 10,000 Markov chain steps. The frequency of the distribution of pairwise differences between sequences (mismatch distribution) [19] and neutrality tests [26, 27]

**Table 1. Accession numbers of nucleotide sequences retrieved in Genbank database and included in this study.**

| Species | Sample origin | Accession numbers | Citation |
|---|---|---|---|
| *Loxodonta cyclotis* | Côte d'Ivoire | KJ557424.1 | [40] |
| | Gabon | KJ557423 | [40] |
| | Sierra Léone | JN673264 | [41] |
| | République centrafricaine | JN673263 | [41] |
| *Loxodonta africana* | Savanne 1 | AB443879.1 | [42] |
| | Savanne 2 | NC_000934.1 | [43] |
| | Savanne 3 | DQ316069.1 | [44] |
| *Elephas maximus* | Asie 1 | DQ316068 | [44] |
| | Asie 2 | NC_005129 | [44] |
| | Asie 3 | EF588275 | [45] |
| *Mammuthus primigenius* | Mammouth | EU155210 | [46] |

were examined to test the historical changes in effective population size of forest elephants. The neutrality of the mutations in the compared sequences was estimated with the software Arlequin version 3.5. Pairwise mismatch distribution was tested using the software DnaSP version 6.12.01 [21].

The past population history of forest elephants in Côte d'Ivoire was estimated using Bayesian skyline plot (BSP) using BEAST2 2.6.7 software [31]. The optimal model of evolution was estimated with bModelTest [32]. whilst *"bPopSizes"*, *"bGroupSizes"* and *"Bayesian"* were set to three, and other parameters were left as default. An uniform calibration time prior was applied to the root of the Mammalia clade (79.4–110.4 Myr) [33, 34]. We used a relaxed clock (log normal-distributed) and 100,000,000 MCMC iterations. The initial 10% of the MCMC was discarded as burn-in. Trees and model parameters were sampled every 5,000 generations. We used a mutation rate of 0.05 per million years as the scale factors for the mtDNA and multiplied by 0.5 because only the female mtDNA contributes to the effective population size [35]. For Bayesian skyline plot visualization, the RevBayes v 1.2.1 software [36] was used to regenerate a new analysis matrix from the one created by Beast. This newly obtained matrix is used for *Plotting the output of RevBayes analyses in the R package RevGadgets* [36]. The substitution model chosen for this analysis was Hasegawa, Kishino and Yano (HKY) and applying a strict molecular clock model [37].

To investigate the possible historical demographic scenarios of *Loxodonta cyclotis*, we implemented approximate Bayesian computation (ABC) methods using the program DIYABC v.2.0.4 [38]. We defined two demographic scenarios to determine the possible demographic history of *L. cyclotis* of Côte d'Ivoire following the recommendations of Cabrera and Palsboll [39] (Fig 1). In scenario 1 (S1 model), one population from which several samples have been taken at various generations: 0, 3 and 10. The only unknown parameter of the scenario is the effective population size. By grouping all individuals from these three sub-populations together randomly, in scenario 2 (S2 model), two populations of size N1 and N2 have diverged t generations in the past from an ancestral population of size N1+N2 (Fig 1). We simulated 100 000 000 datasets for each scenario and calculated summary statistics using the software DIYABC DIYABC v.2.1.0 [39]. The posterior probability of each scenario was obtained by a direct approach and by the logistic regression approach, as implemented in DIYABC v.2.1.0 [39].

## Results

### Genetic diversity and haplotype frequency

DNA was successfully obtained from 158 dung samples stored in 20% DMSO buffer. For the total sample collected, the amplification success rate was 86.70% (137 out of 158 samples). Of the 137/158 samples with positive simplex PCR, 80.30% (110 of 137) were successfully sequenced. This success rate was distributed by sample collection site as follows: 100% (51/51), 74.54% (41/55), 52% (13/25) and 75% (6/8) for amplicons obtained from dung samples collected in Bossématié CF, Dassioko CF, Port Gauthier CF and Abidjan Zoo respectively. Of the 137 samples with a positive simplex PCR, 80.30% (110 out of 137) were successfully sequenced. As part of this study, 47, 41 and 13 good quality sequenced sequences of the mitochondrial DNA control region (CSB1) obtained from dung samples respectively collected in the classified forests of Bossématié (FCB), Dassioko (FCD) and Port-Gauthier (FCPG), were taken into account for different analyses.

Sequences fragment of the mitochondrial DNA control region (CSB1) measuring 600 bp were obtained from 101 individuals of the elephant populations sampled in the classified forests. The remainder of the samples could not be used due to the low quality of mtDNA they contained. Then 101 sequences were considered for analyzes of different genetic parameters.

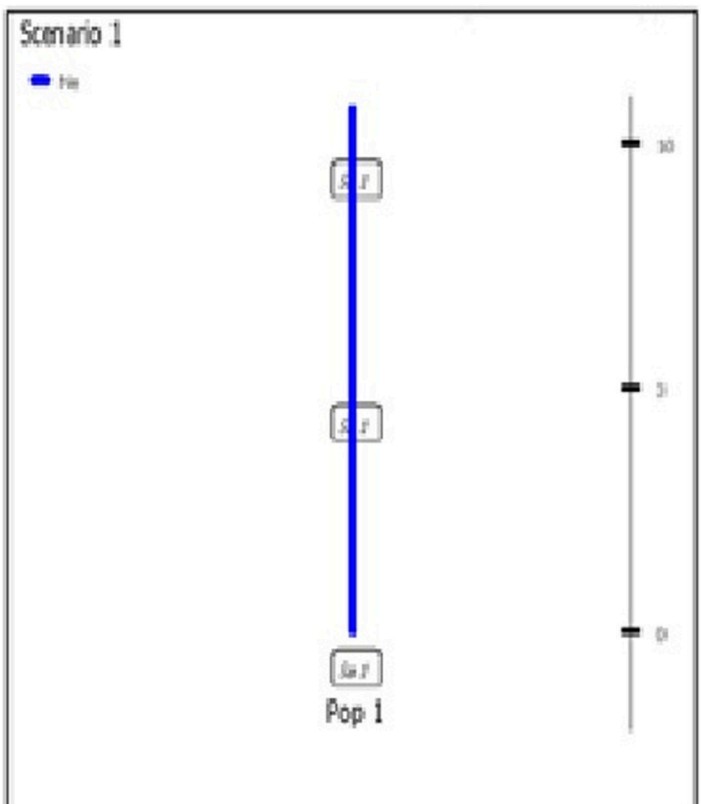 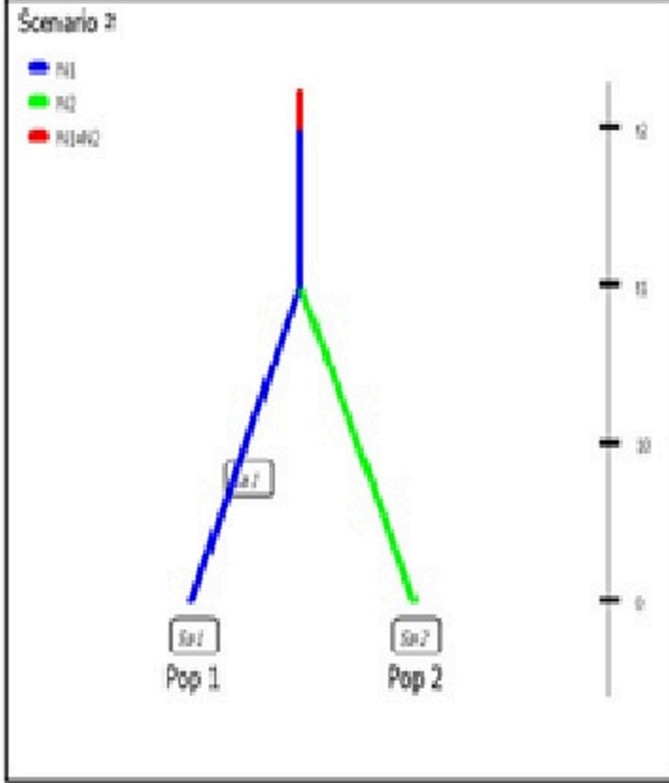

**Fig 1.**

Ten other sequences were deleted because of their poor quality. Our sequences have 95 to 100% identity with the mtDNA sequences of forest elephant existing in GenBank.

The alignment of 101 sequences (600 bp) of the mDNA control region (CSB1) of forest elephants of Côte d'Ivoire yielded 53 polymorphic sites and 26 haplotypes. Haplotype diversity was Hd: 0.863 ± 0.019 and the value of Nucleotide diversity, Pi (Jukes-Cantor): 0.00416. For the entire population analyzed, the most frequent haplotype was Hap11 (23.76%) followed by haplotype Hap1 (22.77%). In the Bossématié CF, with respect to the population of that forest, haplotypes Hap1 had the highest relative frequency (i.e. 48.94%, 23 out of 47 samples) followed by haplotype Hap3 (i.e. 34.04%, 16 out of 47 samples) (Figs 2 and 3).

For the Dassioko CF, the haplotype Hap11 predominated (i.e. 46.34%, 19 out of 41 samples) followed by haplotypes Hap16 (14.63%, 6 out of 41 samples) and Hap20 (12.19%, 5 out of 41 samples (Figs 2 and 3). In the Port Gauthier CF, the haplotype Hap11 predominated (38.46%, or 5 out of 13 samples), followed by haplotypes Hap16 and Hap23 (15.40%, or 2 out of 13 samples) (Figs 2 and 3). Fifteen (15) out of 26 haplotypes were singletons. No haplotype was shared between the Bossématié CF and the other two sites (Dassioko and Port Gauthier CFs) whereas these two latest sites shared 5 haplotypes (Figs 2 and 3). The detail on the frequency, distribution of the identified haplotypes is indicated on Table 2. Haplotype Hap3 is shared between the Bossématié CF and the elephant kept in the zoo of Abidjan. Although the Bossématié CF did not share any haplotype with the Dassioko and Port Gauthier FR, the delineation between some of these haplotypes is weak. For example, only one mutational step separates the haplotype Hap3 (unique to Bossématié) to haplotype Hap11 co-occurring in Dassioko and Port Gauthier FRs (Figs 2 and 3).

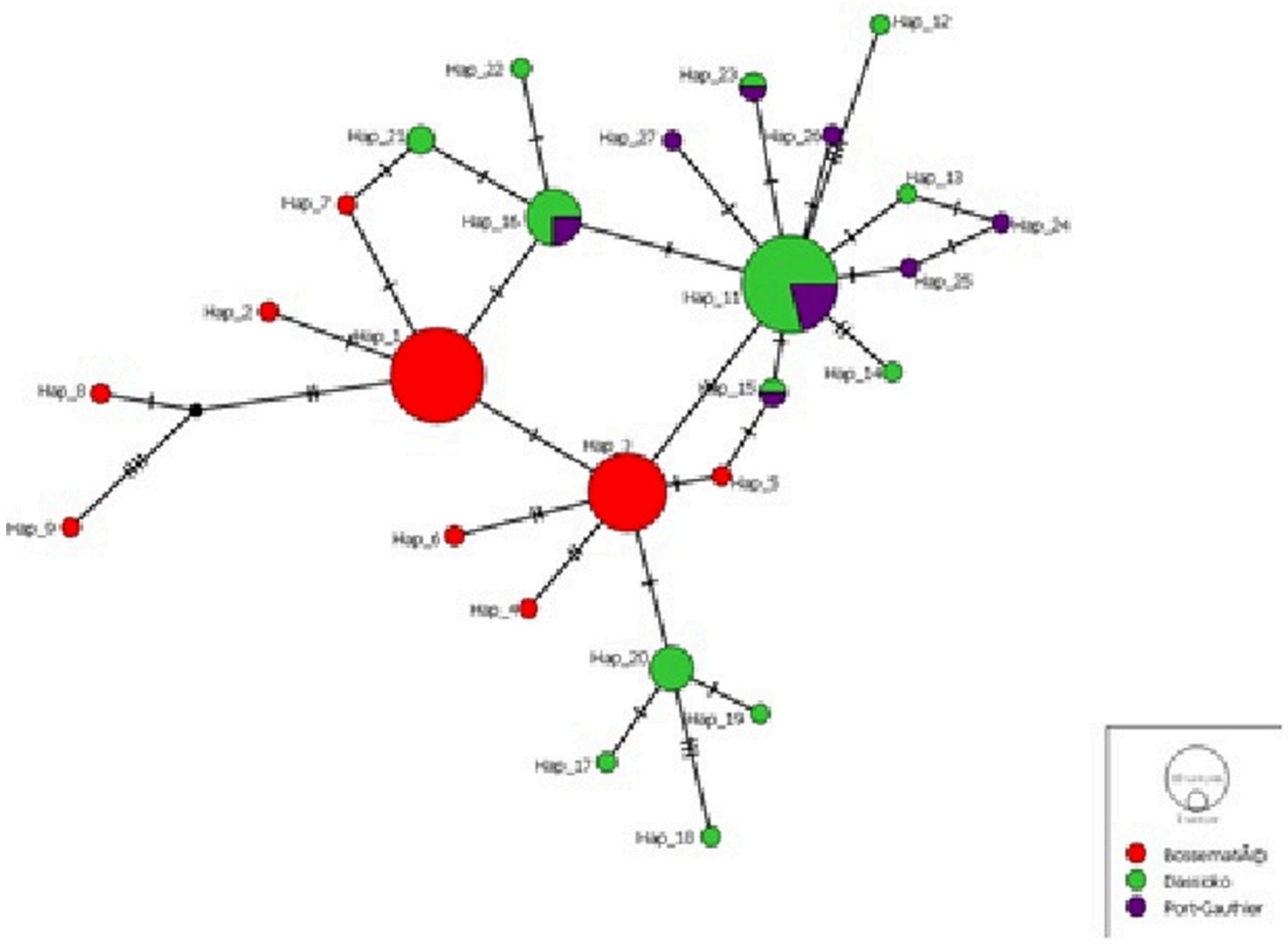

**Fig 2.**

The haplotype Hap_3 detected in the elephants of Bossématié FC and that of the Abidjan Zoo was identical to the haplotype found within the population of forest elephant in the Taï national park in Côte d'Ivoire (KJ557424.1). In exception for the haplotype Hap3, the other 26 haplotypes did not match to any haplotype in the public database.

## Population structure

The populations of forest elephants considered as a whole exhibited strong differentiation ($F_{ST}$ = 0.265, Snn = 0.806, p-value = 0.00). When comparing the three subpopulations, significant genetic differentiation was observed either between the subpopulation of Bossématié and Dassioko $F_{ST}$ = 0.271, p-value = 0.0) or Bossématié and Port Gauthier CFs ($F_{ST}$ = 0.375, p-value = 0.0). A low genetic differentiation was observed between Dassioko and Port Gauthier FRs ($F_{ST}$ = 0.03, p-value = 0.22), suggesting the existence of genetic exchanges between these populations.

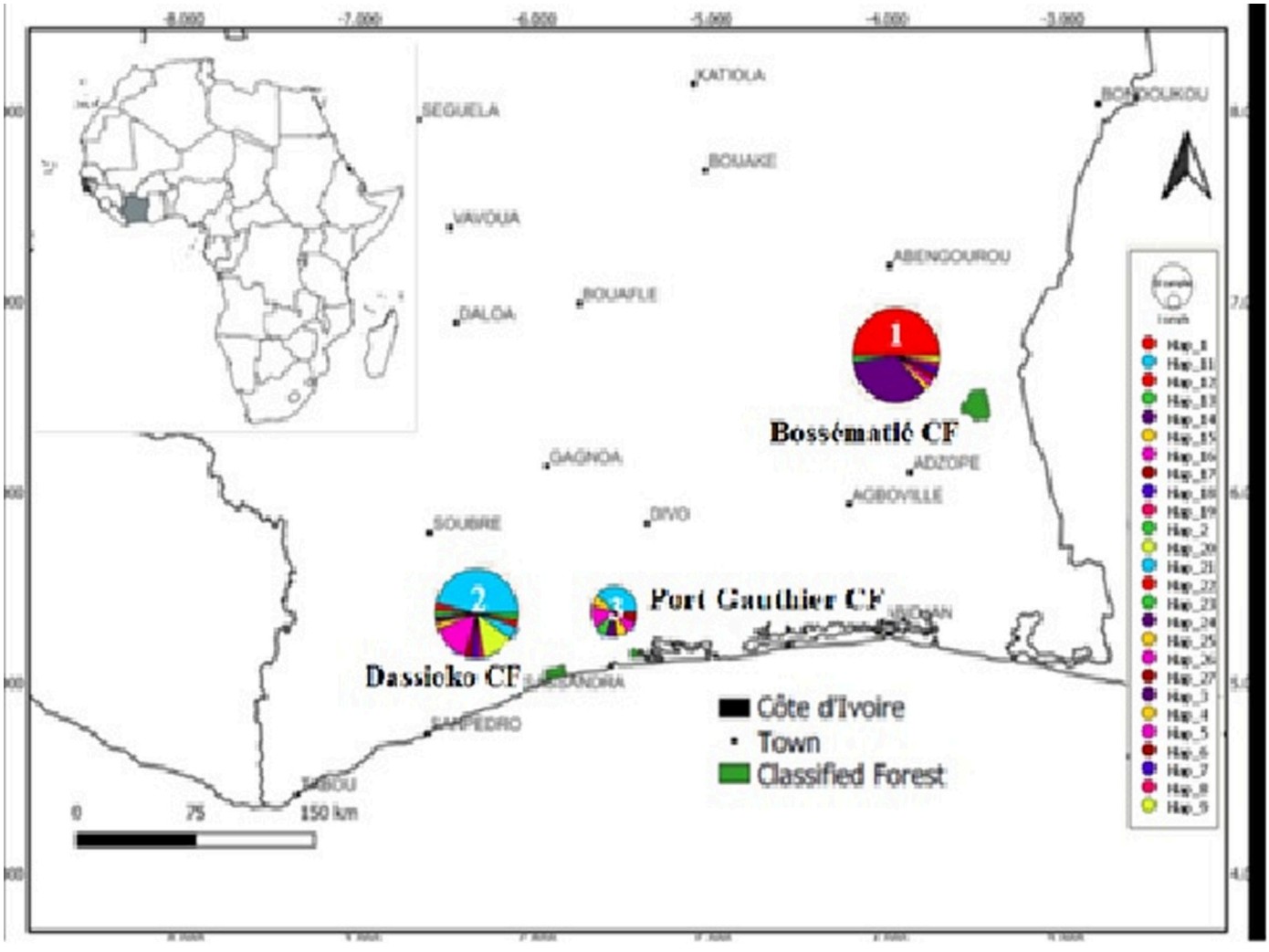

**Fig 3.**

When grouping the populations according to their geographic origins, the analysis of molecular variance (AMOVA) indicated that the majority of the total genetic variance (73.55%) occurred within populations and to a lesser extent among populations (26.45%). A highly significant differentiation was observed within populations ($F_{ST}$ = 0.265, p = 0 < 0.05) (Table 2).

The Mantel test shows a positive but not significant correlation, between genetic and geographical distances (Monte Carlo randomisation test, Rxy = 0.97; P = 0.37). (Fig 4).

**Table 2. Analyses of molecular variance (AMOVA) in *Loxodonta cyclotis* in Côte d'Ivoire inferred from mitochondria control region.**

| Source of variation | Sum of squares | Variance component | Pourcentage of variation (%) | F-statistics | P-values |
|---|---|---|---|---|---|
| Three populations based on their geographic origin | | | | | |
| Among populations | 24.475 | 0.36875 | 26.45 | | |
| Within populations | 100.475 | 1.02526 | 73.55 | 0.265 | < 0.0001 |
| Total | 124.950 | 139401 | 100 | | |

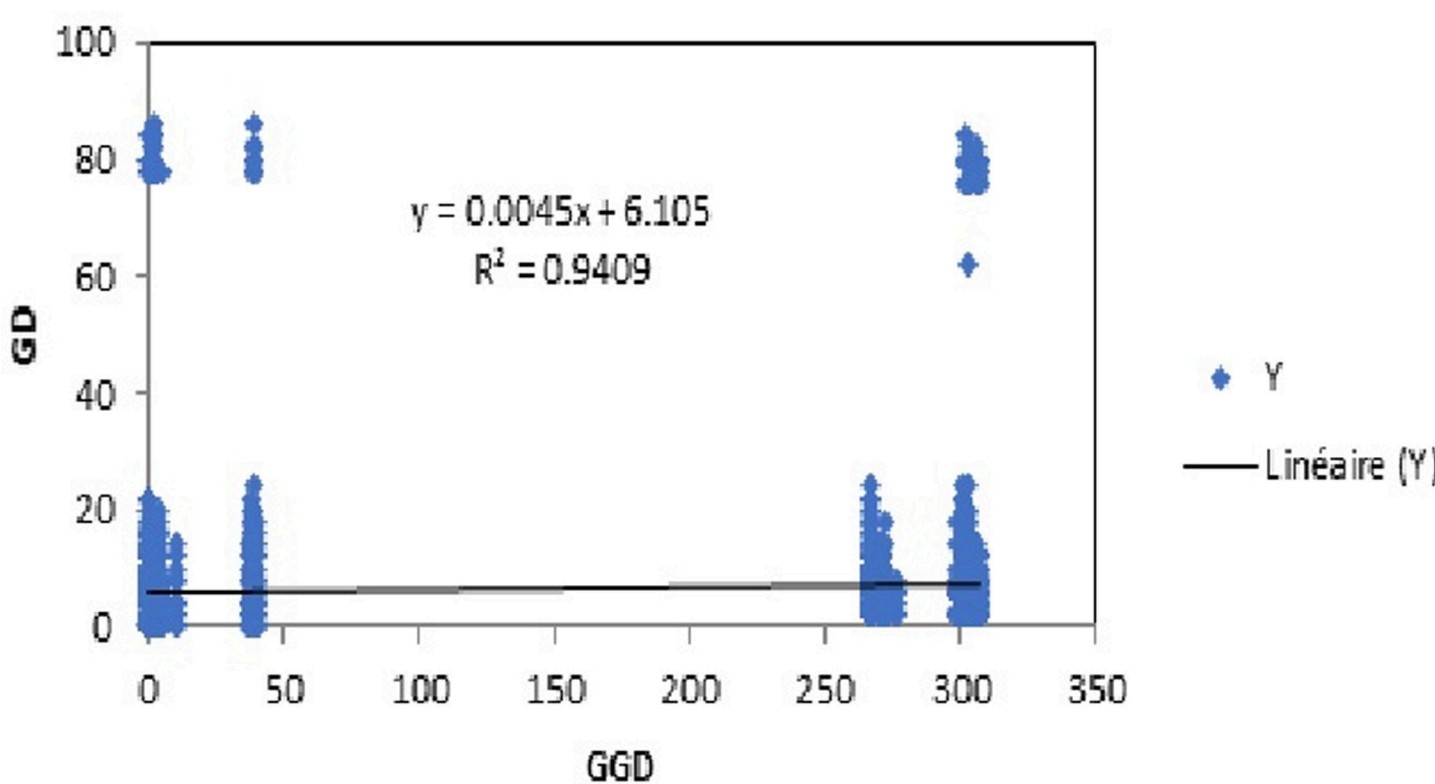

**Fig 4.**

Fig 5 illustrates the relationships between the mtDNA control region sequences generated during this study and those of forest (*Loxodonta cyclotis*), savanna (*L. africana*), *Mammuthus primigenius*, and *Elephas maximus* retrieved from the National Center for Biotechnology Information (NCBI) database.

The phylogenetic analysis of the subpopulations of the Bossématié, Dassioko, and Port Gauthier CFs shows that they appear monophyletic, with heterogeneous structuring (genetic variation) in a clade with the forest elephants selected in the NCBI database (Fig 5).

From the sequence analyzed, the phylogenetic study showed that forest elephants and savanna elephants are not two distinct species but rather forms mixed populations. This analysis revealed a concordant deep genetic divergence between African forest and savanna elephant populations and those of Asian elephant (*Elephas maximus*) and Woolly mammoth *Mammuthus primigenius* (Fig 5). On the other hand, the latter two are phylogenetically close (Fig 5).

## Historical demography

Neutrality tests, either by the Fs value of Fu and Li and the D value of Tajima's are negative (Table 3). These tests are however significant for the population of Bossématié CF (Tajima's D = -2.481, p = 0.0, Fu = -1.251, p = 0.31) and Dassioko CF (Tajima's D = -1.845, p = 0.017, Fu = -6.115, p = 0.002) and not significant for the population of Port Gauthier CF according to the Tajima's Test (Tajima's D = -1.55, p = 0.052) (Table 3). On the other hand, according to the Fu and Li Test, the neutrality test was significant for the populations of Dassioko CF (Fs = -6.115,

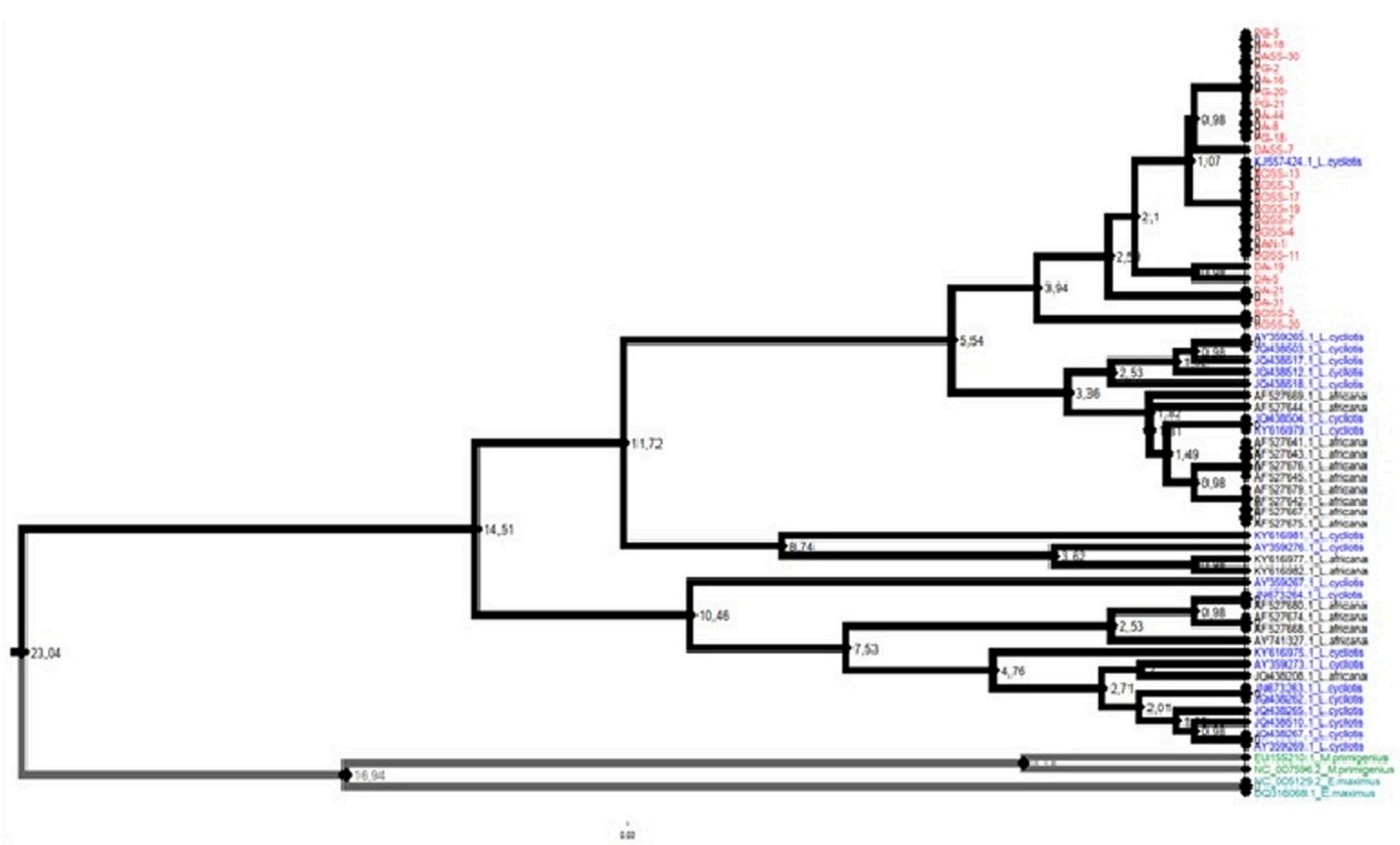

**Fig 5.**

p = 0,002) and Port Gauthier CF (Fs = -5.332, p = 0.0) and not for the one of Bossématié CF (Fs = -1.251, p = 0.31) (Table 3).

Bayesian methods were used in order to calibrate elephant samples, producing an overall mitochondrial mutation rate of 0.05 and a Higher Posterior Distribution Interval (HPDI) of -1096.2371 and -935.2983. The HPDI were rather narrow, with a well-resolved peak in the sampling.

The graphical Bayesian Skyline plot (BSP) represents the inferred demographic trajectory (Fig 6). It consists of a plot with three curves, representing the point estimates (median) and 95% HPD intervals of θ through time. The most significant net diversification occurred over the period from 2.8 to 1.8 million years ago. However, from 1 to 0 million years ago net diversification remained constant (Fig 6). The BSP analysis shows that the greatest expansion occurred between 2.8 and 2.6 million years ago. From 2.6 to 2 million years ago, there was a

**Table 3. Genetic diversity and neutrality tests of the mtDNA control region (CSB1) of forest elephant of Côte d'Ivoire.**

| Populations | Number of sequences | Number of haplotypes | Haplotype diversity (Hd) | Tajima's test (D) | Fu's Fs (FS) test | Significance index (p-value) | | Raggedness index (r) |
|---|---|---|---|---|---|---|---|---|
| | | | | | | Tajima's | Fu's Fs | |
| FCB | 47 | 10 | 0.655 ± 0.050 | -2.481 | -1.251 | 0.0 | 0.31 | r = 0.12 ; p> 0.05 |
| FCD | 41 | 13 | 0.759 ± 0.062 | -1.845 | -6.115 | 0.017 | 0.002 | r = 0.021 ; p> 0.05 |
| FCPG | 13 | 8 | 0.859 ± 0.088 | -1.55 | -5.332 | 0.052 | 0.0 | r = 0.12 ; p> 0.05 |

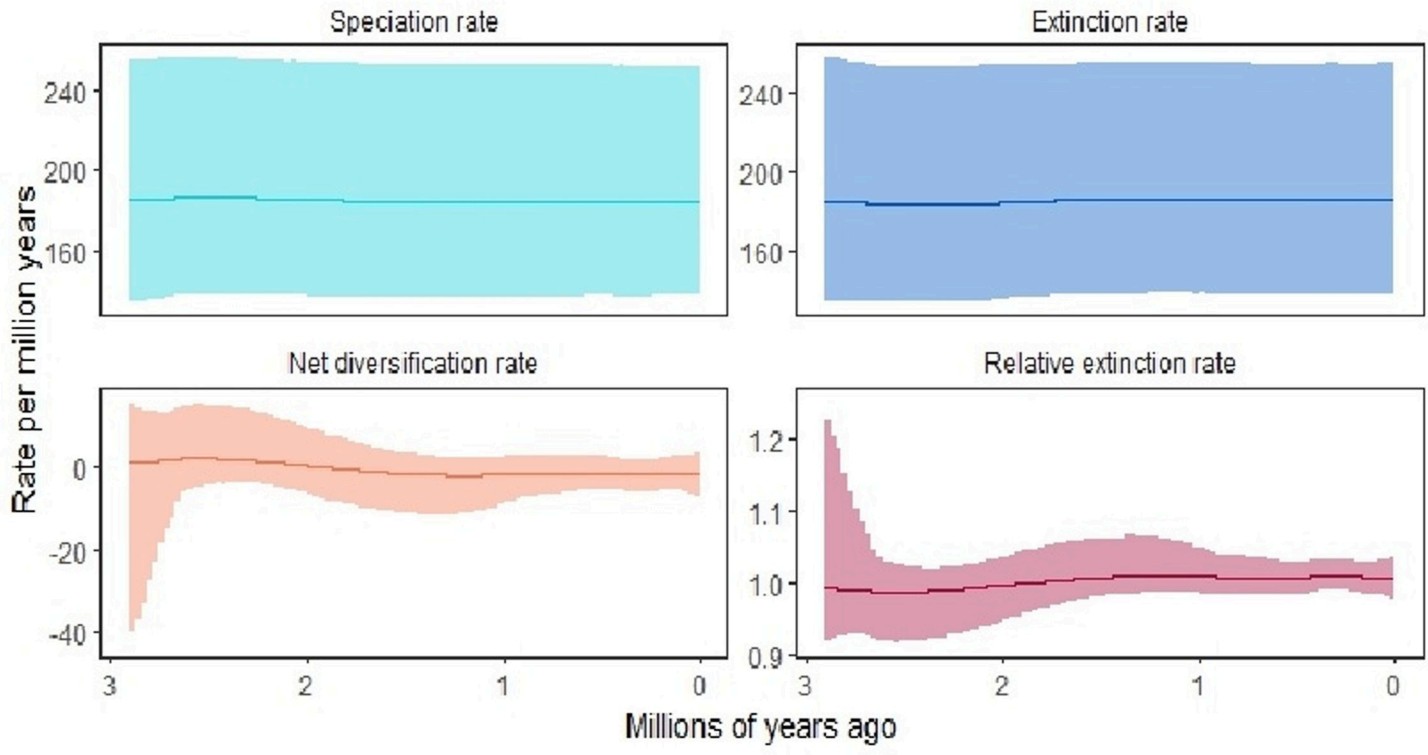

**Fig 6.**

regression of the elephant population. But, from 2 to 1 million years ago there was again a magnitude of expansion in elephant population size and it remained constant from 1 to 0 million years ago (Fig 6).

On the ABC modeling of the possible demographic history of *Loxodonta cyclotis* of Côte d'Ivoire, the "S1 model" was highly favored (posterior probability = 0.8718 (0.0000, 1.0000)) over the "S2 model" (posterior probability = 0.7889 (0.0010, 1.0000)). The data simulated according to the population growth models indicate an unimodal curve with respective values of R2 of 0.052 and 0.087 within Dassioko CF and Port Gauthier CF populations respectively. In addition, the data simulated according to the demographic expansion model gives a multimodal distribution with a value of R2 = 0.09 obtained at within the Bossématié CF population.

These results showed a demographic change in these subpopulations, indicating that the Bossématié CF elephant population exhibit greater population fluctuation compared with those of Dassioko and Port Gautier CFs. Furthermore, these demographic parameters show that elephant population of Bossématié CF has experienced recent population expansion, unlike those of the Dassioko and Port Gauthier FRs.

However, the Raggedness statistic (r), was not significant, and therefore did not support the hypothesis of the expansion of forest elephant populations in the studied areas: Bossématié CF (r = 0.12, p > 0.05), Dassioko CF (r = 0.021, p > 0.05) and Port Gauthier CF (r = 0.12, p> 0.05) (Table 3). Thus, suggesting that these populations have remained stable.

## Discussion

This study provided the first wide examination of the population genetic structure of forest elephants in Côte d'Ivoire. An average nucleotide diversity of 3.5% was observed in the

mitochondrial DNA control region of the studied populations. This value is higher than that reported by Nyakaana et al. [47] within savannah elephants in Zimbabwe and by Fernando et al. [48] within Asian elephants (1.8%). It is however lower than these reported in several studies using the same marker; for example, in Grant's gazelle (6.2%) [49], and African buffalo (5.0%) [50]. Relative to the low level of genetic diversity observed, haplotypic diversity was high (0.757). The low level of nucleotide diversity and the high level of haplotypic diversity generally indicate a small effective population size and could also be the result of founder effect and bottlenecks [10, 51].

More than half (16 of 31) of the haplotypes observed were singletons and only the Dassioko and Port Gauthier CFs shared the same haplotypes (Hap11, Hap15, Hap16 and Hap23). Since these haplotypes follow the evolutionary system of maternal sex influence [52, 53], these exchanges of gene flows indicate their ancestral maternal origins [12, 54, 55], and a strong connectivity of maternal gene flow between elephant populations in these two forests. The haplotype Hap 3 detected in the elephants of Bossématié CF and in the Abidjan zoo is identical to the haplotype found in the forest elephant of the Taï national park in Côte d'Ivoire (KJ557424.1) identified by previous studies. However, the haplotypes detected in Côte d'Ivoire Forest elephants were all different from these detected in other African countries. This is in favor of a local-specific distribution of forest elephant haplotype across tropical Africa. In the same vein, Ishida et al. [56] sampled from 22 locations in Africa and found 20 locations in which elephants carried location-specific haplotypes not detected among elephants from other locations.

Furthermore, our study shows that gene flow occurs between the two populations of Dassioko CF and Port Gauthier CF, unlike the population of Bossématié CF which did not share any haplotype with these two populations. The Dassioko and Port Gauthier CFs are located in the same region (Gboklê) and are separated by 41 km. The proximity of these two forests and the possible connecting corridors between them enable gene flow between elephant populations.

The high gene flows between these two populations certainly decreased or prevented the geographic differentiation between the whole populations studied. Indeed, the results of the Mantel test indicated a positive but not a significant correlation, between the genetic distance and geographic distance amongst the mtDNA sub-populations, which does not suggest isolation by distance(IBD) [57, 58].

This structure differs from the genetic structuring described for elephant populations in Tanzania where structuring is ascribed to anthropogenic and landscape [59] and the founder effects among the elephants of the Kavanga-Zambezi area [60].

The ability of elephants to travel long distances and to adapt to various ecological niches has certainly affected such a structuring of the populations studied. This lack of structuring is also supported by the phylogenetic tree and the haplotype network.

Our study indicated that a country-specific distribution of haplotypes and the haplotype network showed a star-shaped topology with a high rate of singletons. This reflects a low geographical dispersion of the mtDNA due to the fact that the females do not migrate between herds of elephants [61].

These assertions are in contradiction with the strong genetic divergence separating the Bossématié population from those of Dassioko and Port Gauthier. This disproportionality could be explained by several factors. Indeed, although a relatively large geographical distance separates Dassioko and Port Gauthier CFs with that of Bossématié CF (about 250 km), it does not constitute a barrier for elephants. However, the strong human induced disturbance (plantations, human settlements, roads, . . .) of the lands separating these forests could limit genetic exchanges between elephant populations. Indeed, elephant's movements are not made at random but are determined by various factors including human activities [62]. Thus, the high

levels of fragmentation of protected areas where elephants have survived strongly contributed to the limitation of gene flow when there is no direct proximity that can promote gene exchanges, show by the Dassioko and Port Gauthier CFs. This is supported by the AMOVA analysis and the Snn statistic [63] which showed significant genetic differentiation between the studied populations. Habitat fragmentation has led to the reproductive isolation of the elephant populations studied, with a more pronounced effect between the populations of the Bossématié CF and those of Dassioko and Port Gauthier CFs based on the non-significance of the Raggedness statistic (r).

The genetic profile established in this study is based on mitochondrial DNA (mtDNA) control region, which is only transmitted maternally. This non-coding genetic marker is commonly used in conservation genetic studies to examine genetic structure at the population level because it is considered to be the most rapidly evolving region of the mitochondrial genome [64–66]. This locus is broadly considered to be suitable for examining intraspecific relationships among closely related groups of individuals [67]. However, the use of mtDNA control region presents some limitations resulting from the fact that the analysis of mtDNA corresponds to the study of a single locus. This means that the total dependence on mtDNA as a marker in the study of the natural history of populations will give only a small part of that history [68]. Indeed, the high evolutionary rate that has made the control region an attractive marker for biologists may be masking the true relationships between populations due to high haplotype diversity as well as homoplasy. Hence, the use of an alternative marker with a slower evolutionary rate is needed to highlight the population structure of forest elephant.

The demographic history of the studied populations indicates that they have remained stable during their evolutionary history. However, despite the relatively high genetic diversity indices observed in these populations, the strong decrease in their size could affect their survival in the near future, in the absence of gene flows between them. The interpretation of the results of the neutrality tests used could be biased if the assumption that the sample is taken from a single randomly mating population is violated. The high levels of intra-population and inter-population genetic diversity observed indicate that there was a relatively large elephant population in these different forests. A significant portion of the old gene pools seems to be still maintained in the present populations despite the large reduction in their size. This is an encouraging result for the conservation of this critically endangered species. Despite recent disturbances caused by poaching pressure and the fragmentation of the habitats of these pachyderms, their populations have maintained a sufficient level of genetic diversity. Indeed, the fragmentation of the habitat of these animals leads to their isolation into subpopulations, thus reducing gene flow [69].

The mutation rate calculated in the present was 0.05 (HPDI -1096.2371 and -935.2983), which differs from those obtained by Barnes et al. [70], Bonillas-Monge [71] and Fu et al. [72], which are 1.48E-06, 3.13E-07, and 2.53E-08, respectively in Woolly Mammoth, Leopard Seal, and Human.

The ability of the ABC skyline graph to detect changes in population size varied widely between the different scenarios evaluated. However, demographic changes of small magnitude and close to the present were the hardest to detect has reported in previous studies [73, 74].

But, inferred demographic trajectory was good for constant size populations. The two supported scenarios (1 and 2) both displayed admixed populations elephants in the three classified forests. In scenario 2, the ancestral populations of lineages I colonized these three sub-populations (FCD, FCB and FCPG) (Fig 1).

The results obtained are similar to other previous molecular studies showing that African forest and savanna elephants are phylogenetically closer than those of Asia and to the woolly mammoth. However, the latter species are much more closely related to each other [40, 42, 45,

75]. The mtDNA phylogenetic tree obtained was non-monophyletic for forest and savannah elephant species as previously reported by several studies [76–78]. Additional insight into the population dynamics of the population of forest elephant in Côte d'Ivoire, based on nuclear markers will be of paramount importance.

We are aware that Phylogenetic relationships cannot be reliably inferred using only one variable genetic marker. Particularly data from nonrecombining mitochondrial DNA (mtDNA), has important limitations that are now widely recognized. Indeed, a mtDNA tree may be different from the population or species tree due to the effects of natural selection [79, 80], introgression [81] or the wide stochastic variance that characterizes a sample of gene trees collected from a set of populations or species [82]. Hence, no definitive conclusions about their phylogenetic relationships can be drawn from the obtained data alone.

### Implication for conservation

Forest elephant populations have sharply declined in their size and number over the past decades and the species is now Critically Endangered. The survival of the species now constitutes a great challenge for conservationists. Indeed, the main objective of the conservation of an endangered species is to increase the effective size of its population by maintaining gene flows and its overall genetic diversity. Any strategy for the sustainable conservation of forest elephant populations should therefore integrate the genetic approach.

Our study has shown that elephant populations of the Dassioko, Port Gauthier and Bossématié CFs share the same evolutionary lineage as indicated by the phylogenetic tree. The sharing of several haplotypes between the populations of Dassioko and Port Gauthier showed the existence of an ancestral genetic corridor between these two populations as opposed to the population of Bossématié which remains isolated. However, the genetic corridor connecting the Dassioko and Port Gauthier CFs remains seriously threatened given the current presence of human activity (plantations, human settlements and road infrastructures) all along that corridor. The long-term survival of these two sub-populations will therefore depend on the maintenance and restoration of that corridor, given the small size of the remaining populations. The situation remains more worrying for the forest elephant population of the Bossématié CF which, in addition to its small size, remains geographically isolated. A genetic corridor should therefore be established for the benefit of that population. In Côte d'Ivoire, the Bossématié CF is located at the vicinity of several other protected areas such as Yaya and Mabi FRs. Unfortunately, forest elephant has been locally extirpated from these forests. A genetic corridor should be established on the border of neighboring Ghana where there are several protected areas (Krokosua National Park, Boin NP) inhabiting forest elephants that are located near the Bossématié FR. Indeed, maintaining genetic connectivity between protected areas is crucial for the long-term survival of species [83].

Recent surveys by Kouakou et al. [5] reported the presence of forest elephants in the Taï National Park and of some isolated individuals in communities forests. A complete profile of the genetic structure of forest elephants in Côte d'Ivoire should include these populations. The survival of individuals isolated in the vicinity of plantations depends solely on their reintroduction into the populations living in protected areas. In addition to ensuring the survival of these individuals, this reintroduction will genetically enrich the host populations.

### Conclusions

The analysis of the sequences of the mitochondrial DNA control region of forest elephant populations of Bossématié, Dassioko and Port Gauthier CFs indicated a strong genetic connectivity between the last two populations. Our study has revealed a high genetic differentiation

between the populations of Bossématié CF and those of the other two forests, indicating a low gene flow between these subpopulations, probably due to their strong fragmentation and geographic isolation.

A strong ancestral genetic connectivity was found between the populations of Dassioko and Port Gauthier CFs. That connectivity certainly contributed to the relatively high genetic diversity found within these populations and certainly gives them a selective advantage compared with forest elephant population of Bossématié. The viability of this latter population appears very fragile given its isolation and the strong reduction of its size. Urgent measures including the establishment of genetic corridors and the reinforcement of protection measures need to be undertaken to safeguard these remaining populations.

## Supporting information

**S1 File.**
(TXT)

## Acknowledgments

We are grateful to the Minister of Scientific Research of Côte d'Ivoire for the research permit and the Swiss Centre of Scientific Research for the logistical support that they offered. We are also grateful to the SODEFOR (Société de Développement des Forêts) and the OIPR (Office Ivoirien des Parcs et Réserves) for the permit to access the protected areas. We thank the local communities around the surveyed forests and the field guides for their assistance. The study received the approval of the SODEFOR (Société de Développement des Forêts) and the OIPR (Office Ivoirien des Parcs et Réserves).

## Author Contributions

**Conceptualization:** Sery Gonedelé-Bi.

**Data curation:** Sery Gonedelé-Bi.

**Formal analysis:** Jean-Louis Kouakou, Sery Gonedelé-Bi.

**Investigation:** Jean-Louis Kouakou.

**Methodology:** Jean-Louis Kouakou, Sery Gonedelé-Bi.

**Software:** Jean-Louis Kouakou.

**Supervision:** Sery Gonedelé-Bi.

**Validation:** Sery Gonedelé-Bi.

**Writing – original draft:** Jean-Louis Kouakou, Sery Gonedelé-Bi.

**Writing – review & editing:** Jean-Louis Kouakou, Sery Gonedelé-Bi.

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
