## [Decision Letter · Decision Letter 0]

26 Dec 2023

PONE-D-23-19598Population genetic structure and historical demography of the population of forest elephants in Côte d’IvoirePLOS ONE

Dear Dr. Gonedelé Bi,

Thank you for submitting your manuscript to PLOS ONE. After careful consideration, we feel that it has merit but does not fully meet PLOS ONE’s publication criteria as it currently stands. Therefore, we invite you to submit a revised version of the manuscript that addresses the points raised during the review process.

We look forward to receiving your revised manuscript.

Kind regards,

Tunira Bhadauria, Ph.D.

Academic Editor

PLOS ONE

Journal Requirements:

Reviewers' comments:

Reviewer's Responses to Questions

**Comments to the Author**

1. Is the manuscript technically sound, and do the data support the conclusions?

Reviewer #1: Partly

Reviewer #2: Yes

2. Has the statistical analysis been performed appropriately and rigorously? 

Reviewer #1: Yes

Reviewer #2: Yes

3. Have the authors made all data underlying the findings in their manuscript fully available?

Reviewer #1: Yes

Reviewer #2: Yes

4. Is the manuscript presented in an intelligible fashion and written in standard English?

Reviewer #1: Yes

Reviewer #2: Yes

5. Review Comments to the Author

Reviewer #1: The manuscript analyzes 101 sequences from three populations of Loxodonta cyclotis in Côte d’Ivoire (Africa): Bossématié, Dassioko and Port-Gauthier classified forests. The authors analyzed a 600pb fragment from the hypervariable I section of the mitochondrial control region. Based on this genetic marker they evaluated the genetic diversity, the population genetic structure, and population demographic history of forest elephant in Côte d’Ivoire. The authors observed low gene flow between the Bossématié population and the other populations, likely resulting from geographical isolation caused by fragmentation. Additionally, they found that the size of Loxodonta cyclotis populations in Côte d’Ivoire remained constant over time.

This manuscript represents a high contribution towards the Loxodonta cyclotis genetic population and conservation. However, I have noticed some crucial issues that need to be addressed:

1) The text requires reorganization. For example, the methodology should clearly state how the sequences were curated leaving only 101 sequences and avoid re-presenting information already presented in the result section in the discussion section.

2) The utilization of a highly variable sequence to infer species diversity and population historical demography represents a weakness in this study. The authors should, at the very least, acknowledge their awareness of this limitation. In comparison, all other studies used for reference employed at least two genetic markers, including less variable regions. If the authors followed a similar approach, they should emphasize it in the manuscript.

3) I noticed that the supplementary material mentioned in line 326 is missing, as well as the GenBank accession numbers of the sequences they obtained.

4) Figure 4 and line 383: Based on this graphic, it is not possible to infer a correlation between genetic distance and geographical distance. While it is conceivable that this genetic marker should exhibit an isolation by distance pattern, the current graphic does not demonstrate such a relationship.

5) Phylogenetic relationships cannot be reliably inferred using only one variable genetic marker. I recommend refraining from drawing such conclusions throughout the text. As for Figure 5, it is evident that these individuals possess distinct haplotypes in comparison to other works, while also sharing similar haplotypes among themselves. However, no definitive conclusions about their phylogenetic relationships can be drawn from this data alone.

Reviewer #2: This is well-written manuscript and based on impressive empirical evidence and makes an original contribution. The abstract is adequately addressed. The Introduction is adequate. The literature review is concise updated covering the main relevant concept and empirical studies on different approaches on this field. In material methods sample collection, DNA extraction, PCR sequencing and data analysis is properly maintained as per this research work demand . The results are clearly presented with proper statistical methods and adequately addressed. The authors develop a unique theoretical framework and highlighted originality much more. The study may be assist to maintain elephant populations of Dassioko, Port Gauthier, Zoo of Abidjan and Bossematie . Therefore , the manuscript can be considered for the publication.

6. PLOS authors have the option to publish the peer review history of their article (what does this mean?). If published, this will include your full peer review and any attached files.

Reviewer #1: **Yes: **Juliana Cordeiro

Reviewer #2: **Yes: **Dr. Dhananjay Singh

---

## [Author Response · Author response to Decision Letter 0]

29 Jan 2024

Reviewer #1: 

This manuscript represents a high contribution towards the Loxodonta cyclotis genetic population and conservation. However, I have noticed some crucial issues that need to be addressed:

1) The text requires reorganization. For example, the methodology should clearly state how the sequences were curated leaving only 101 sequences and avoid re-presenting information already presented in the result section in the discussion section.

1) Answer

Of the 137 samples with a positive simplex PCR, 80.30 % (110 out of 137) were successfully sequenced. As part of this study, 47, 41 and 13 good quality sequences of the mitochondrial DNA control region (CSB1) obtained from dung samples respectively collected in the Bossématié (FCB), Dassioko (FCD) and Port-Gauthier (FCPG) Forest Reserves. These 130 sequences were taken into account for different analyses. 

2) The utilization of a highly variable sequence to infer species diversity and population historical demography represents a weakness in this study. The authors should, at the very least, acknowledge their awareness of this limitation. In comparison, all other studies used for reference employed at least two genetic markers, including less variable regions. If the authors followed a similar approach, they should emphasize it in the manuscript.

2) Answer

The genetic profile established in this study is based on mitochondrial DNA (mtDNA) control region, which is only transmitted maternally. This non-coding genetic marker commonly used in conservation genetic studies to examine genetic structure at the population level because it is considered to be the most rapidly evolving region of the mitochondrial genome (Baker and Marshall 1997; Ruokonen 2002). This locus is broadly considered to be suitable for examining intraspecific relationships among closely related groups of individuals (Sinclair and Pe´rez-Losada 2005). However, the use of mtDNA control region presents some limitations resulting from the fact that the analysis of mtDNA corresponds to the study of a single locus. This means that the total dependence on mtDNA as a marker in the study of the natural history of populations will give only a small part of that history (Zhang & Hewitt 2003). Indeed, the high evolutionary rate that has made the control region an attractive marker for biologists may be masking the true relationships between populations due to high haplotype diversity as well as homoplasy. Hence, the use of an alternative marker with a slower evolutionary rate may be more suitable than the control region to reveal the population structure of forest elephant.

3) I noticed that the supplementary material mentioned in line 326 is missing, as well as the GenBank accession numbers of the sequences they obtained.

3) Answer

The GenBank accession numbers of the sequences obtained have been included as suggested by the reviewer. The additional material mentioned is available from line 521 to 523.

Data Availability Statement: Sequence data are available in GenBank https://www.ncbi.nlm.nih.gov/nuccore/MZ485910.1,MZ485851.1,MZ485850.1,

MZ485849.1,MZ485848.1,MZ485847.1,MZ485846.1,MZ485845.1,MZ485844.1,MZ485843.1,MZ485842.1,MZ485841.1,MZ485840.1,MZ485839.1,MZ485838.1,MZ485834.1,MZ485833.1,KJ557424.1,MZ485831.1,MZ485830.1,MZ485829.1,MZ485828.1,MZ485827.1,MZ485826.1,MZ485825.1,MZ485824.1,MZ485823.1,MZ485822.1,MZ485821.1,MZ485820.1,MZ485819.1,MZ485818.1,MZ485817.1,MZ485816.1,MZ485815.1,MZ485814.1,MZ485813.1,MZ485812.1,MZ485811.1,MZ485810.1,MZ485809.1,MZ485852.1,MZ485909.1,MZ485903.1,MZ485900.1,MZ485899.1,MZ485898.1,MZ485896.1,MZ485895.1,MZ485894.1,MZ485892.1,MZ485891.1,MZ485890.1,MZ485889.1,MZ485881.1,MZ485880.1,MZ485879.1,MZ485878.1,MZ485877.1,MZ485871.1,MZ485870.1,MZ485869.1,MZ485866.1,MZ485865.1,MZ485864.1,MZ485863.1,MZ485862.1,MZ485861.1,MZ485859.1,MZ485858.1,MZ485856.1,JQ438209.1,MZ485905.1,MZ485904.1,MZ485887.1,MZ485886.1,MZ485885.1,MZ485882.1,MZ485873.1,MZ485872.1,MZ485836.1,MZ485832.1,MZ485888.1,MZ485883.1,MZ485908.1,MZ485907.1,MZ485906.1,MZ485902.1,MZ485876.1,MZ485874.1,MZ485860.1,MZ485837.1,MZ485835.1,JQ438745.1,MZ485893.1,MZ485884.1,MZ485901.1,MZ485897.1,MZ485868.1,MZ485867.1

List the number of samples collected in the Genbank database used in this study (Tables 3).

Table 3 Accession numbers for samples included in this study.

Species Sample origin Accession number Citation

Loxodonta cyclotis Côte d’Ivoire KJ557424.1 [71]

 Gabon KJ557423 [71]

 Sierra Léone JN673264 [70]

 République centrafricaine JN673263 [70]

Loxodonta africana Savanne 1 AB443879.1 [80] 

 Savanne 2 NC_000934.1 [81]

 Savanne 3 DQ316069.1 [82]

Elephas maximus Asie 1 DQ316068 [73]

 Asie 2 NC_005129 [73]

 Asie 3 EF588275 [64]

Mammuthus primigenius Mammouth EU155210 [83]

4) Figure 4 and line 383: Based on this graphic, it is not possible to infer a correlation between genetic distance and geographical distance. While it is conceivable that this genetic marker should exhibit an isolation by distance pattern, the current graphic does not demonstrate such a relationship.

4) Answer

The Mantel test shows a strong and positive but not significant correlation, between genetic and geographical distances (Monte Carlo randomisation test, Rxy = 0.97; P = 0.37). Which does not suggest isolation by distance (IBD). 

5) Phylogenetic relationships cannot be reliably inferred using only one variable genetic marker. I recommend refraining from drawing such conclusions throughout the text. As for Figure 5, it is evident that these individuals possess distinct haplotypes in comparison to other works, while also sharing similar haplotypes among themselves. However, no definitive conclusions about their phylogenetic relationships can be drawn from this data alone.

5) Answer

Figure 5 illustrates the relationships between the mtDNA control region sequences generated during this study and those of forest elephant (Loxodonta cyclotis), savanna elephant (L. africana), Mammuthus primigenius, and Elephas maximus retrieved from the National Center for Biotechnology Information (NCBI) database. The phylogenetic analysis of the subpopulations of the Bossématié, Dassioko, and Port Gauthier CFs shows that they appear monophyletic, with heterogeneous structuring (genetic variation) in a clade with the forest elephants selected in the NCBI da-tabase. Based on the sequence analysed, the phylogenetic study showed that forest elephants and savannah elephants form two different clades. This analysis detected that Asian elephants (E. maximus) and the Mammuthus primigenius clade form a clade in their own right, which differs from those of forest elephants and savannah elephants (Fig 5).

We are aware that Phylogenetic relationships cannot be reliably inferred using only one variable genetic marker. Particularly data from nonrecombining mitochondrial DNA (mtDNA), has important limitations that ares now widely recognized. Indeed, a mtDNA tree may be different from the population or species tree due to the effects of natural selection (see, for example, Weinreich & Rand 2000 and Bazin et al. 2006), introgression (Chan & Levin 2005; Melo-Ferreira et al. 2005) or the wide stochastic variance that characterizes a sample of gene trees collected from a set of populations or species (Hey & Machado 2003). Hence, no definitive conclusions about their phylogenetic relationships can be drawn from the obtained data alone.

Reviewer #2: 

This is well-written manuscript and based on impressive empirical evidence and makes an original contribution. The abstract is adequately addressed. The Introduction is adequate. The literature review is concise updated covering the main relevant concept and empirical studies on different approaches on this field. In material methods sample collection, DNA extraction, PCR sequencing and data analysis is properly maintained as per this research work demand. The results are clearly presented with proper statistical methods and adequately addressed. The authors develop a unique theoretical framework and highlighted originality much more. The study may be assist to maintain elephant populations of Dassioko, Port Gauthier, Zoo of Abidjan and Bossematie. Therefore, the manuscript can be considered for the publication.

Good

---

## [Editor Report · Decision Letter 1]

28 Feb 2024

Population genetic structure and historical demography of the population of forest elephants in Côte d’Ivoire

PONE-D-23-19598R1

Dear Dr. Bi

We’re pleased to inform you that your manuscript has been judged scientifically suitable for publication and will be formally accepted for publication once it meets all outstanding technical requirements.

Kind regards,

Tunira Bhadauria, Ph.D.

Academic Editor

PLOS ONE
---

## [Editor Report · Acceptance letter]

4 Apr 2024

PONE-D-23-19598R1 

PLOS ONE

Dear Dr. Gonedelé Bi, 

I'm pleased to inform you that your manuscript has been deemed suitable for publication in PLOS ONE. Congratulations! Your manuscript is now being handed over to our production team.

Kind regards, 

on behalf of

Dr. Tunira Bhadauria 

Academic Editor

PLOS ONE